# Bioeffectors as Biotechnological Tools to Boost Plant Innate Immunity: Signal Transduction Pathways Involved

**DOI:** 10.3390/plants9121731

**Published:** 2020-12-08

**Authors:** Helena Martin-Rivilla, Ana Garcia-Villaraco, Beatriz Ramos-Solano, Francisco Javier Gutierrez-Mañero, Jose Antonio Lucas

**Affiliations:** Plant Physiology, Pharmaceutical and Health Sciences Department, Faculty of Pharmacy, Universidad San Pablo-CEU Universities, 28668 Boadilla del Monte, Spain; anabec.fcex@ceu.es (A.G.-V.); bramsol@ceu.es (B.R.-S.); jgutierrez.fcex@ceu.es (F.J.G.-M.); alucgar@ceu.es (J.A.L.)

**Keywords:** bioeffector, beneficial rhizobacteria, metabolic elicitors, induced systemic resistance (ISR), *Pseudomonas syringae* pv. *tomato* DC3000, SA and JA/ET signal transduction pathways

## Abstract

The use of beneficial rhizobacteria (bioeffectors) and their derived metabolic elicitors are efficient biotechnological alternatives in plant immune system elicitation. This work aimed to check the ability of 25 bacterial strains isolated from the rhizosphere of *Nicotiana glauca*, and selected for their biochemical traits from a group of 175, to trigger the innate immune system of *Arabidopsis thaliana* seedlings against the pathogen *Pseudomonas syringae* pv. *tomato* DC3000. The five strains more effective in preventing pathogen infection were used to elucidate signal transduction pathways involved in the plant immune response by studying the differential expression of Salicylic acid and Jasmonic acid/Ethylene pathway marker genes. Some strains stimulated both pathways, while others stimulated either one or the other. The metabolic elicitors of two strains, chosen for the differential expression results of the genes studied, were extracted using n-hexane, ethyl acetate, and n-butanol, and their capacity to mimic bacterial effect to trigger the plant immune system was studied. N-hexane and ethyl acetate were the most effective fractions against the pathogen in both strains, achieving similar protection rates although gene expression responses were different from that obtained by the bacteria. These results open an amount of biotechnological possibilities to develop biological products for agriculture.

## 1. Introduction

The diseases caused by different pathogen organisms in plants represent a significant and persistent threat and a challenge to supply food worldwide [1,2]. Because of that, the study of plants’ immune system as a mechanism to counteract the attack of pathogens is fundamental, especially in this year that has been declared International Year of Plant Health by the FAO (Food and Agriculture Organization of the United Nations).

Plants can activate patter-triggered-immunity (PTI) by the recognition of PAMPs/MAMPs (pathogen-/microbe-associated molecular patterns), or effector-triggered immunity [3] (ETI) by the recognition of pathogen effectors. PTI response activates when some specific receptors located on cells surface, called pattern recognition receptors (PRRs), detect these PAMPs/MAMPs. However, plants can also respond to endogenous molecules that have been released by pathogens, which implies recognition of virulent pathogen molecules, called effectors, by intracellular receptors. This last recognition leads to a second line of defense, the effector-triggered immunity (ETI) and also to the transcription of resistance genes (PR genes). These endogenous effectors recognized by plants are much more variable in structure and composition than PAMPs/MAMPs [4].

Pieterse et al. [5] classified induced resistance triggered by pathogens with respect to the type of triggering agent in: Systemic acquired resistance (SAR), herbivore induced resistance (HIR) and induced systemic resistance (ISR). SAR is a form of induced resistance that happens in plants after localized exposure to a pathogen and that depends on the accumulation of salicylic acid (SA) and the activation of the Nonexpressor of Pathogenesis-Related Protein 1 (NPR1). SA accumulates after pathogen infection, binding NPR1 and triggering induction of pathogenesis-related genes (PR). Although SA-mediated resistance acts against a wide plethora of pathogens, it has been reported that SAR is generally more effective facing to biotrophic and hemibiotrophic pathogens [6,7].

In contrast, Pieterse et al. [8] described ISR as an answer triggered by non-pathogen rhizobacteria (bioeffectors). However, different elicitors such as antibiotics, surfactants or chemical inducers [9] are also able to induce ISR. In this case, ISR response was described as dependent on jasmonic acid (JA) and ethylene (ET) signaling pathways and also needs the involvement of NPR1 [10,11]. Plant defensin1 (*PDF1*) [12,13], and *MYC2* also play an essential role in this signaling pathway [14].

These bioeffectors and some of their elicitors (structural molecules or metabolic molecules released to the medium) induce in plants a physiological alert state prior to stress challenge known as priming [15]. Plants in this state are able to develop a faster and/or stronger activation of defensive responses after the attack of pathogens, insects or in response to abiotic stress [16]. After bioeffectors or their elicitors are sensed, the SA or JA/ET signaling pathways are activated to trigger plant resistance [17]. Therefore, the study of these transduction signal pathways is meaningful for understanding the plant immune system and their defenses against pathogens. This can contribute to promote the use of bioeffectors and their elicitors as a useful biotechnological strategy to develop a sustainable agriculture without using agrochemicals and pesticides [17].

It is known that the rhizosphere of wild plant species is a worthy source for finding putative effective beneficial bacteria (bioeffectors) because plants are able to select those bacteria that boost their fitness by releasing nutrients into the surrounding soil through root exudates [18,19]. Thanks to this well-known ability of the plants to strongly select beneficial bacterial strains in the rhizosphere to improve their health and to survive to adverse conditions [18,19,20], bacteria isolated from the rhizosphere of *Nicotiana glauca* Graham, a *Solanaceae* native of Southern Spain with a strong secondary metabolism [21], were studied. This plant was chosen for our work because it is a wild species capable of thriving in very poor soils and in extreme drought and temperature conditions, therefore it was deduced that it should have a powerful associated rhizospheric microbiome that would allow it to survive to these harsh conditions. Moreover, *N. glauca* synthesizes anabasin, a toxic alkaloid, which suggests the existence of an inducible secondary metabolism associated to defence mechanisms [21]. All these characteristics of this plant species led us to presume that the rhizosphere of wild populations of *N. glauca* would be a rich source for finding effective bioeffectors with very interesting metabolic capacities.

The effects induced in the plants by the beneficial rhizobacteria depend on molecules (elicitors), so we considered that after the extraction of these elicitors, it would be possible to find out which ones were able to reproduce the effects of the rhizobacteria and therefore were responsible of this effect.

The general objective of this work was therefore to find beneficial rhizobacteria (bioeffectors) from *N. glauca* rhizosphere efficient in triggering the innate defense response of *A. thaliana* plants, as well as effective derived metabolic elicitors, trying to elucidate the mechanisms involved in the protection. To achieve this objective the following partial objectives were defined: (i) To perform a screening of *N. glauca* rhizobacteria to select those strains efficient in triggering the innate response of *Arabidopsis* plants against the pathogen *P. syringae* DC3000, (ii) to study the mechanisms involved in plant defense triggered by the most effective bioeffectors against the pathogen *P. syringae* DC3000, (iii) to obtain the metabolic elicitors from the most effective bioeffectors and assay their ability to mimic bacterial response.

To reach our goals, ISR experiments were carried out in *A. thaliana* plants using the bioeffectors and the metabolic elicitors of the chosen strains to protect the plants against *P. syringae* DC3000; the differential expression of marker genes for the SA and JA/ET transduction pathways were studied on plants inoculated with selected strains and selected metabolic elicitors.

The ISR experiments were carried out in *A. thaliana* plants because its short life cycle facilitates the performance of experiments at laboratory level. Furthermore, it is easy to infect this plant with specific pathogens (e.g., *P. syringae* DC 3000) and to study the consequent plant immune responses. This plant is able to trigger a wide sort of general PTI and ETI responses against pathogens [22]. In addition, it exists extensive literature on the defensive responses of this plant to compare with our results. As it is a model plant, the data obtained would be very relevant for the scientific community and could be then extrapolated to crops of commercial or economic interest.

## 2. Results

### 2.1. Beneficial Rhizobacteria Screening: Phylogenetic Tree and Biochemical Tests

A phylogenetic tree was performed with the 16S rRNA sequences of the 175 bacterial strains (Figure 1). Two main groups appeared, one made up of Gram-positive (74 strains) and the other of Gram-negative bacteria (101 strains).

In the Gram-negative group, eight genera were found (*Serratia*, *Enterobacter*, *Pantoea*, *Erwinia*, *Cronobacter*, *Acinetobacter*, *Pseudomonas,* and *Stenotrophomonas*), being *Pseudomonas* especially diverse in species (five species identified: *P. putida*, *P. reinekei*, *P. brassicacearum*, *P. fragi,* and *P. fluorescens*). In the Gram-positive group, only two genera were found, (*Bacillus* and *Brevibacterium*). Within *Bacillus*, two species were especially abundant, *Bacillus cereus* and *Bacillus megaterium* (Figure 1).

Biochemical tests (auxin-like compounds production [23], siderophores production [24], phosphate solubilization [25], and chitinases production [26,27]) for identifying putative beneficial rhizobacteria were carried out to the 175 strains. The results of these tests are shown in Table 1.

Within Gram-negative bacteria, *Enterobacter* was the only genus across all isolates tested that were capable of producing indole acetic acid (IAA). Siderophore producing isolates were present in all genera. *Acinetobacter* and *Pseudomonas* showed the highest percentage of phosphate solubilizes, but also isolates of *Enterobacter*, *Pantoea* and *Erwinia* were able to solubilize phosphate. Finally, all *Stenotrophomonas* isolates were able to produce chitinases (100%). Isolates able to produce siderophores and also solubilize phosphates belonged to *Enterobacter*, *Pantoea, Erwinia, Acinetobacter,* and *Pseudomonas*. Those able to produce siderophores and also chitinases were present among *Stenotrophomonas* and *Pseudomonas*, although less abundant among the latter (2.08%). The unique genus that had isolates with three biochemical traits was *Enterobacter*. It was able to produce siderophores and IAA and also, to solubilize phosphate.

Within Gram-positive bacteria, none of the isolates produced IAA, however all were able to produce siderophores. Only *B. cereus, Brevibacterium* sp., and *B. megaterium* were able to solubilize phosphate. *B. cereus* and *B. subtilis* were able to produce chitinases. The isolates that were able to produce siderophores and also solubilize phosphates were *B. cereus*, *Brevibacterium* sp., and *B. megaterium.* The isolates that were able to produce siderophores and also chitinases were *B. cereus* and *B. subtilis.* The unique isolate that had three biochemical traits was *B. cereus*. It was able to produce siderophores and chitinases and also to solubilize phosphate.

### 2.2. ISR by Beneficial Rhizobacteria

According to the results obtained from the phylogenetic tree (Figure 1) and the biochemical tests (Table 1), 25 strains were chosen (15 Gram-negative and 10 Gram-positive) to develop a first protection experiment against the pathogen *P. syringae* DC3000. All selected strains had at least two or three biochemical traits, except N 10.7 *Serratia odorifera*, N 12.34 *S. rubidaea,* and N 11.14 *Bacillus endophyticus* that only had one activity, but they were able to reduce growth of other strains in plate (data not shown), probably due to the production of antibiotics. The selected strains and their biochemical traits are shown in Table 2.

Table 3 shows the percentage (%) of protection induced in seedlings of *A. thaliana* inoculated with the 25 selected strains and the percentage of protection of negative and positive control plants. All Gram-negative bacteria significantly protected against the pathogen, except N 8.22, N 10.6, N 10.21, N 15.23, and N 18.10. Protection achieved by N 16.24 was not statistically significant. N 5.12 (*P. putida*), N 8.17 (*S. maltophilia*), N 12.34 (*S. rubidaea*), and N 21.24 (*P. fluorescens*) were the Gram-negative bacteria that induced the highest protection, even above of that of the positive control. Therefore, these four strains were chosen for assessing differential gene expression of eight genes, markers of different signal transduction pathways related to plant immune system.

Within Gram-positive bacteria, all of them significantly protected against the pathogen, except N 11.14, N 11.22, and N 11.36. Strain N 4.1 (*B. cereus*) was the Gram-positive bacterium that performed best; hence, it was selected to assess the differential gene expression of eight genes, markers of different signal transduction pathways related to plant immune system.

Differential gene expression at 6, 12, and 24 h after pathogen challenge (hapc) of *A. thaliana* plants inoculated with selected strains (N 5.12 (*P. putida*), N 8.17 (*S. maltophilia*), N 12.34 (*S. rubidaea*), N 21.24 (*P. fluorescens*), and N 4.1 (*B. cereus*) is shown in Figure 2, Figure 3, Figure 4, Figure 5 and Figure 6. Three different behaviors appeared among the five strains. The first behavior was a strong and significant increase at 6 hapc, followed by strains N 5.12 (Figure 2) and N 21.24 (Figure 5); N 5.12 increased the expression of *NPR1* (12.55 times), *PDF1* (376.54 times) and *PR3* (4.53 times), while N 21.24 strongly induced *ICS* (42.11 times) and *LOX2* (10.66 times) at 6 hapc. A second behavior pattern was a significant increase in expression at 12 hapc, only followed by N 12.34 (Figure 4) with a very high increment of the differential expression of *NPR1*(149.74 times), *PR2* (57.09 times), *PDF1* (675.98 times), *PR3* (41.37 times), and *LOX2* (32.79 times). The third pattern was a significant increase at 24 hapc, followed by strains N 8.17 (Figure 3) and N 4.1 (Figure 6). *ICS* (1.79 times), *PR1* (1.95 times), *PR2* (2.22 times), and *MYC2* (2.02 times) were the genes induced by N 8.17, while all genes studied were induced by N 4.1 (from 1.24 times for *MYC2* until 5.01 times for *NPR1*).

### 2.3. ISR by Metabolic Elicitors

Based on all the previous results, two strains were selected to extract their metabolic elicitors and to check the capacity of these metabolic elicitors to mimic protective effects of bacteria. They were selected N 12.34 (*S. rubidaea*), the Gram-negative strain that showed the highest differential expression (Figure 4) and N 4.1 (*B. cereus*) as it was the Gram-positive strain with better protection among the Gram-positive and which ranked second among all (Table 3).

The three fractions extracted from each strain (n-hexane, ethyl acetate and n-butanol), achieved significant protection (Table 4), having an outstanding performance the metabolic elicitors in the n-hexane and ethyl acetate fractions. Protection of the n-hexane (61.26%) and the ethyl acetate (54.64%) fractions of N 12.34 and protection of the n-hexane (68.11%) and the ethyl acetate (67.30%) fractions of N 4.1 was similar to that obtained with the bacterial strains (56.64% for N 12.34 and 69.45% for N 4.1, respectively).

Differential gene expression induced by the metabolic elicitors in the n-hexane and ethyl acetate fractions (the fractions with greatest protective capacity) from N 12.34 and N 4.1 is shown in Figure 7. In the case of N 12.34, analysis was performed at 6 and 12 hapc, and in that of N 4.1, at 12 and 24 hapc. Genes and sampling moments were selected according to the results obtained in the previous experiment of differential gene expression (with the bioeffectors).

The two metabolic elicitor fractions from N 12.34 induced the same behavior in the genes studied: Expression of *NPR1* and *PR2* increased from 6 to 12 hapc, while *PDF1* decreased. Both metabolic elicitor fractions from N 4.1 also had the same behavior: Expression of *NPR1* and *PDF1* decreased from 12 to 24 hapc, while *PR3* increased.

## 3. Discussion

In the present study, the efficiency of bioeffectors and derived metabolic elicitors to trigger the immune system of *A. thaliana* conferring protection against *P. syringae* pv. *tomato* DC3000 has been shown.

The 175 strains were isolated in 2010 [21] from the rhizosphere of wild populations of *N. glauca*. This plant species was chosen as it was hypothesized that its very active secondary metabolism would select a good group of bacteria to ensure plant fitness.

The rationale of plant’s selection capacity has been widely demonstrated, and also the use of the rhizosphere as a source of highly specialized strains [29,30,31], since it is one of the most complex and diverse ecosystems on earth. This suggests a definite role of plant-derived metabolites in the microbiome assemblage in the rhizosphere [32,33]. According to previous results, the common culturable bacterial genera in the rhizosphere of *N. glauca* includes *Bacillus* sp., *Pseudomonas* sp., *Enterobacter* sp., *Acinetobacter* sp., *Burkholderia* sp., *Arthrobacter* sp., and *Paenibacillus* sp. [21].

In the present study, almost 100% of the strains produced siderophores. Siderophore production is related to iron limiting nutrient [29,34,35], but also has been related to biocontrol and/or systemic induction of secondary metabolism. Therefore, siderophore-producing strains may have the ability to protect plants against pathogens through a complex and inducible secondary metabolism, which is probably related to defense [36,37].

Regarding the production of auxins and the ability to solubilize insoluble phosphorus, only one genus of those of our study was capable of producing auxins (*Enterobacter* sp). However, the solubilization of phosphates was a very abundant activity among the strains studied. Our results support that *N. glauca* selects rhizobacteria related to nutrition or biocontrol activities (phosphate solubilization and siderophore production) rather than those able to affect plant growth regulator balance (auxins production).

The production of chitinases was well represented within the Gram-positive group, but among the Gram-negatives, only the *Stenotrophomonas* genus was able to produce them, consistent with Ramos Solano et al. [21]. Many species of rhizospheric microorganisms produce chitinolytic enzymes to protect themselves against fungi, since chitin is a major structural component of most fungal cell walls. Hence, these microorganisms have an excellent potential as biocontrol agents [38,39].

The strains that were selected for ISR experiment were able to produce siderophores, and they had also some other complementary capacities, mainly the production of chitinases. This selection criterion has already been used by other authors with the aim of finding bacteria capable of inducing systemic resistance in plants [21,40,41,42]. The strain N 16.15 (*Enterobacter* sp.) was the only non-siderophore producing isolate, but it was one of the two strains that produced auxins, and was chosen for this reason. Some authors have shown that auxins are related to the induction of systemic resistance [43,44]. Three strains, N 10.7 (*S. odorifera*), N 12.34 (*S. rubidaea*) and N 11.14 (*B. enterophyticus*) were chosen with only one biochemical trait, because of their capacity to reduce growth of other strains in plate (data not shown), probably due to the production of antibiotics. This working scheme has proved to be very effective, since 16 out of the 25 strains chosen induced systemic resistance against the pathogen *P. syringae* DC3000 (Table 3).

To determine signal transduction pathways triggered by the five outstanding strains, from the 25 previously selected, the differential expression of marker genes of the SA and JA/ET signaling pathways was studied. For this experiment, the criterion followed for the bioeffector selection was the highest protection against *P. syringae* DC 3000 infection within both bacterial groups (Gram-positive and Gram-negative). To date, most bioeffectors studied for their ability to trigger ISR mechanisms belong to the group of Gram-negative bacteria, especially bacteria of the genus *Pseudomonas* [45]. However, Gram-positive bacteria, and among them, those of the genus *Bacillus*, have gained much importance in the last decade because of the great potential to trigger resistance mechanisms against a wide range of pathogens [46,47].

Three types of defensive responses were detected, according to the time needed to increase gene expression: Rapid, intermediate and slow. The rapid response (at 6 hapc) was generated by strains N 5.12 (*P. putida*) (Figure 2) and N 21.24 (*P. fluorescens*) (Figure 5). N 5.12 induced a strong differential expression of *NPR1*, a marker of SA pathway, *PDF1* and *PR3*, markers of the JA/ET pathway. Interestingly, N 21.24 induced a strong differential expression of *ICS* and *LOX2* involved in SA and JA synthesis, respectively. The intermediate response (at 12 hapc) was produced by N 12.34 (*S. rubidaea*) (Figure 4), which induced a strong differential expression of markers of SA pathway (*NPR1* and *PR2*), and markers of the JA/ET pathway (*PDF1* and *PR3*). The different behavior generated by these three strains is also reflected in their defensive capacity. Although the three induced resistance above the positive control (BTH), N 5.12 and N 12.34 induced a lower protection than N 21.24, which was the most effective of all the tested. Contrary to Caarls et al. [48], we observed a simultaneous high expression of *NPR1* and *PDF1* at 6 hapc for N 5.12 and at 12 hapc for N 12.34, suggesting that SA is not suppressing the expression of *PDF1* as these authors indicated. This may be related to the monomerization process of NPR1 protein, (which has not been determined in this work), as well as with the location of this protein (nucleus or cytoplasm), which plays an important role in the suppression or not of the genes involved in the synthesis of JA by SA [49,50]. The higher protection achieved by N 21.24 (Table 3), is probably related to the high expression of the genes related to the synthesis of SA and JA (*ICS* and *LOX2*) at 6 hapc (Figure 5), something that was specific to this strain. Nowadays, the importance of high concentrations of SA and JA to trigger defensive responses mediated by both hormones is widely accepted [5,48,50]. Slow-response strains showed a progressive increase on expression from 0 to 24 hapc. These strains, N 8.17 (*S. maltophilia*) (Figure 3) and N 4.1 (*B. cereus*) (Figure 6) ranked right after N 21.24 in *Arabidopsis* protection (Table 3). N 8.17 follows the classic SA response pathway elicitation by a beneficial strain: High expression levels of *ICS* and *NPR1* and consequently, high expression levels of *PR1*, while genes related with the JA/ET pathway were not expressed. Strain N 4.1 was able to stimulate both pathways (SA and JA/ET) simultaneously, according to the high expression levels of SA markers genes (*NPR1, ICS,* and *PR1*) and JA/ET markers (*PDF1*, *LOX 2* and *PR3*) (Figure 6), demonstrating again that these two pathways are not necessarily antagonistic, as previously indicated by several authors [51,52].

Based on gene expression and protection results, the Gram-negative *Serratia rubidaea* N 12.34 and the Gram-positive *Bacillus cereus* N 4.1 were selected to extract their metabolic elicitors. Bacterial elicitors capable of starting defensive immune responses in plants, have been found to be structural molecules, (e.g., flagellin [53]), or metabolic elicitors that are released into the medium [17,45,54,55,56,57]. Our research delves into the study of mixtures of metabolic elicitors extracted from rhizobacteria and according to their solubility in three different organic solvents. The objective was to compare the effect of these fractions with that of the bacteria (bioeffectors), looking for similarities or differences in the response. For this reason, the genes studied and the hapc sampling moments in each case were set according to the results obtained with the bacterial strains.

For both bacteria, metabolic elicitors in the n-hexane and the ethyl acetate fractions were as efficient in triggering the defensive response in the plant as the bioeffectors (Table 3 and Table 4). Although a lack of effect of structural elicitors cannot be ruled out, it is evidenced herein that both bacteria are capable of releasing metabolic elicitors with the ability to elicit defensive metabolism in the plant very efficiently. On the other hand, since both fractions have elicitation capacity, it seems that the diversity of elicitors is high. This has also been proven by other authors using the same fractions [28,55,58].

Although metabolic elicitors of the two fractions studied protected to the same extent as the bacteria, the expression of the analyzed genes had different behaviors. The strain N 12.34 induced gene expression levels more intensely (up to 140 times. Figure 4) than metabolic elicitors (Figure 7A,B). The different intensity could be due to either the abundance of elicitors when the bacteria is delivered alive, holding all determinants, as compared to a subset of the same elicitors delivered on fractions, or because the plant is more sensitive to elicitors not present in the n-hexane and ethyl acetate fractions. The large difference in the levels of genetic expression indicates a level of priming also different. It is known that the priming can modify the distribution of energetic resources compromising plant growth in favor of a more production of metabolites involved in defensive response [59,60]. Therefore, in this case the use of metabolic elicitors may have advantages over bioeffectors.

Interestingly, metabolic elicitors in both fractions from *S. rubidaea* N 12.34 were able to activate the SA pathway, increasing the expression of *NPR1* and *PR2* (Figure 7A,B). In both fractions, *PDF1* expression (marker of the JA/ET pathway) decreased, which indicate that the metabolic elicitors present in this fraction were only activating the SA mediated transduction pathway, while the bacterial strain activated both. These results show that the elicitors detected by the plant in both cases have to be different, and so would be the PRRs involved in that response [61].

Regarding the *B. cereus* strain N 4.1, the two metabolic elicitor fractions (Figure 7C,D) did not match the bacterium except for *PR3*, a marker of the JA/ET pathway. These results suggest a lower diversity of effective metabolic elicitors, pointing out a more relevant role of structural elicitors triggering the SA mediated pathway observed with the live strain.

All these results show the great number of possibilities offered by elicitors to trigger the immune system of plants, which opens a plethora of biotechnological solutions to different stress situations. Application of elicitors has many advantages from the agronomic point of view because it is more economical and profitable to conserve a molecule than a live bacterium, which has nutritional and environmental requirements. In addition, the use of elicitors also implies less environmental aware for possible cases of ecological niches competition between edaphic species and also avoids problems of infectious pathogenesis and alterations of the rhizosphere [62,63].

## 4. Material and Methods

A screening of 175 isolates was carried out. Firstly, biochemical tests for putative beneficial rhizobacteria traits were carried out to all isolates. The 16S rRNA partial sequencing of all isolates was analyzed and a phylogenetic tree was performed with these sequences. Twenty-five strains selected based on their biochemical traits and avoiding phylogenetic redundancy were assayed to determine their ability to trigger plant protection (ISR). The most effective strains (five) were studied to understand the mechanisms involved in protection. Finally, metabolic elicitors (molecules released to the medium) were obtained from the two most effective bacteria to demonstrate their ability to mimic the protective response triggered by the live strains.

### 4.1. Origin of Bacteria

Bacteria used in this work were isolated from the rhizosphere of wild populations of *N. glauca* Graham in three different soils and physiological stages of the plant. A total of 960 isolates were obtained and 50% were tested for their putative beneficial rhizobacteria traits, as explained in the work of Ramos-Solano et al. [21]. In the present study, a subset of 175 strains from the non-assayed group of bacteria were used. These isolates and the pathogen *P. syringae* pv. *tomato* DC3000 were maintained in 20% glycerol, frozen at −80 °C and plated to check viability.

### 4.2. 16S rRNA Partial Sequencing Phylogenetic Analysis

Bacteria were identified by 16S rRNA partial sequencing phylogenetic analysis. They were grown in PCA (Plate Count Agar (CONDA)) Petri dishes for 48 h and then in nutrient broth (CONDA) under shaking for 24 h at 28 °C in both cases. DNA was extracted from 1.8 mL of each bacterial culture by using the UltranClean Microbial DNA isolation Kit (Mo Bio, Carlsbad, CA, USA, EE.UU). DNA amount and quality were checked with a Nano Drop 2000 Thermo Scientific.

Each DNA sample was amplified with 16S rRNA universal primers: 1492R (5′TACGGYTACCTTGTTACGACTT3′) and 27F (5′AGAGTTTGATCMTGGCTCAG 3′). Amplification reactions were carried out with 5µL DNA (20 ng µL^−1^), 1 unit of DNA polymerase (Biotools Hotsplit), 0.5 µL of Primer F (30 µM) and 0.5 µL of Primer R (30 µM), 2.5 µL of 10X standard reaction buffer with MgCl_2_ (Biotools), 0.625 µL of dNTPs (10 mM each, Biotools), 0.375 µL of 100% DMSO (Dimehyl sulfoxide) and ultrapure water up to a volume of 25 µL.

The reaction mixtures were incubated in a thermocycler (Gene Amp PCR system 2700, Applied Biosystems, South San Francisco, CA, USA) at 94 °C for 2 min and then subjected to 10 cycles, consisting of 94 °C for 0.3 min, 50 °C for 0.30 min and 72 °C for 1 min and 20 cycles consisting of 94 °C for 0.3 min, 50 °C for 0.3 min and 72 °C for 1 min. Finally, the mixtures were incubated at 72 °C for 7 min. PCR products were purified with UltraClean PCR Clean-up DNA purification kit (MO BIO). Purified PCR products were sequenced in an ABI PRIMS” 377 DNA Sequencer (Applied Biosystems). Sequences were visualized with Sequence Scanner software v1.0. (Applied Bio-systems, Foster City, CA, USA), and editing was performed using the software Clone Manager Professional Suite v6.0. (Sci-Ed Software, Cary, NC, USA). Sequence alignment was carried out on the server MAFFT v6.0 (http://mafft.cbrc.jp/alignment/software/) and annotated by BLASTN 2.2.6. in the National Centre for Biotechnology Information (NCBI: http://www.ncbi.nlm.nih.gov/) and Ribosomal Database Project Release 10 (RDP: http://rdp.cme.msu.edu/) databases. Finally, a phylogenetic tree was performed with the 16S rRNA sequences. The sequences reported in this work are available in the GenBank database under the accession numbers, MH571489 to MH571661.

### 4.3. Phylogenetic Tree

An unrooted tree was performed with MEGA v4.0.2. with aligned sequences in MAFFT v6. The evolutionary distances were inferred using the neighbor-joining method. The bootstrap consensus tree inferred from 1000 replicates was taken to represent the evolutionary history of the taxa analyzed. The percentage of replicate trees in which the associated taxa clustered together in more than 50% of the 1000 replicates of the bootstrap test are shown next to the branches. All positions containing gaps and missing data were eliminated from the data set (complete deletion option).

### 4.4. Biochemical Tests for Putative Beneficial Rhizobacteria Traits

The following biochemical tests for putative beneficial rhizobacteria traits were performed on all bacterial isolates: Auxin-like compounds production [23], siderophores production [24], phosphate solubilization [25], and chitinases production [26,27].

For the detection of auxin-like substances, a colorimetric technique was used. Bacterial isolates were inoculated onto half-strength Tryptic Soy Agar (TSA; Difco Laboratories, Sparks, MD, USA) in a grid pattern. Each inoculated plate was overlaid with an 82-mm-diameter nitrocellulose membrane (Amersham Biosciences, Little Chalfont, UK) and subsequently, Salkowski’s reagent (2% (*v/v*) 0.5 M FeCl3 in 35% (*v/v*) perchloric acid) was applied to the membrane and incubated for 2 h to allow for color reaction development. Sensitivity of the assay was determined using known concentrations of pure IAA (Sigma Chemical Co., St. Louis, MO, USA).

To detect the production of siderophores, the culture medium described by Alexander and Zuberer [24] was used. This medium contains CAS (Chrome azurol S), to which iron is bound as part of the Fe-CAS complex, blue in color; those bacteria capable of producing siderophores separate the iron from the complex producing a yellow halo around the bacterial growth zone, a halo that is measured in mm after 24 h of incubation at 28 °C.

To demonstrate the ability of the strains to solubilize phosphate, the bacteria were cultured in the medium described by De Freitas et al. [25], which contains potato dextrose agar and yeast extract (PDYA) with freshly precipitated calcium phosphate. The hydrolysis halo made in this culture medium around the colonies was measured in mm after 24 h of incubation at 28 °C. The presence of a hydrolysis halo confirms the ability of the strain to solubilize phosphate.

To assess the ability of the strains to produce chitinases, the culture medium described by Frändberg and Shunürer et al. [27] was used. This medium contains colloidal chitin, K_2_HPO_4_, KH_2_PO_4_, MgSO_4_, NaCl, KCl, yeast extract and agar. Those bacteria able to produce chitinases produce a transparent halo (the medium is opaque) around the bacterial growth zone, a halo that is measured in mm after 5 days of incubation at 30 °C.

### 4.5. First ISR Experiment. Screening for Isolates Able to Induce Systemic Resistance

Based on phylogenetic analysis and putative beneficial rhizobacteria traits, 25 strains were selected for a first induced systemic resistance (ISR) assay. These bacteria (bioeffectors) were inoculated in *A. thaliana* plants at root level and challenged with the pathogen to evaluate their ability to protect plants.

*A. thaliana* wild type *Columbia ecotype* 0 seeds (provided by the Nottingham Arabidopsis Stock Centre (NASC)) were germinated in quartz sand and two-week-old seedlings were then individually transplanted to 100 mL pots filled with 12:5 (*v/v*) peat/sand mixture (60 g/pot). Forty-eight plants per treatment (strains and controls) were used; plants were arranged in three replicates, with sixteen repetitions each. Plants were watered with 5 mL of tap water once a week and with 5 mL of half-strength Hoagland solution per plant once a week. Strains were inoculated twice by soil drench with 3 mL of a suspension of bacterial cells, grown for 24 h in nutrient broth (CONDA) at 28 °C, and adjusted to a density of 10^8^ cfu mL^−1^, in the first and the second week after transplant. Negative control plants were mock-inoculated by soil drench with 3 mL of sterile nutrient broth and positive control plants were inoculated by soil drench with 10 µL of BTH (Benzothiadiazole) 0.5 mM [28]. Four days after the second bacterial inoculation, plants were pathogen challenged with *P. syringae* DC3000. One day before pathogen challenge, plants were maintained with 99% relative humidity to ensure stomata opening in order to allow disease progress. *P. syringae* DC3000 was centrifuged (10 min at 2890× *g*) and cells were suspended in 10 mM MgSO_4_ to achieve 10^8^ cfu mL^−1^. It was inoculated by spraying the total of the plants with 250 mL. Plants were incubated in a culture chamber (Sanyo MLR-350H) with an 8 h light (350 μE s^−1^ m^−2^ at 24 °C) and 16 h dark period (20 °C) at 70% relative humidity for 72 h, and disease severity was recorded as the number of leaves with disease symptoms relative to the total number of leaves. Results were relativized using the disease severity of negative control plants as 0% protection. All the ISR experimental design is represented as a timeline in Figure 8.

### 4.6. Second ISR Experiment. Study of the Signal Transduction Pathway Involved in Plant Protection

Based on results obtained from the first ISR experiment, the most protective strains (five) were selected to perform a second experiment to analyze the signal transduction pathways involved in plant protection triggered by the bioeffectors. The expression of some marker genes after pathogen challenge were assessed by qPCR. Genes analyzed were *NPR1* (Nonexpressor of Pathogenesis Related Genes1), *PR1* (Pathogenesis-Related Gene 1) and *ICS* (Isochorismate Synthase 1) as markers of the SA signaling pathway [5,48,64,65,66,67,68,69,70]; *PDF1* (Plant Defensin 1), *LOX2* (Lipoxygenase 2) and the transcriptional factor *MYC2* as markers of the JA-ET signaling pathway [39,48,66,68,71,72,73]; and two pathogenesis-related proteins genes, *PR2* (encoding β-1,3-glucanase) and *PR3* (encoding chitinase), as SA and JA/ET markers, respectively [17,50,74,75,76,77,78].

*A. thaliana* was handled as described in the first ISR assay (Figure 8). Instead of recording disease severity 72 hapc, all the leaves of sixteen plants (treated with each bacteria (five)) were harvested at 6, 12 and 24 hapc, powdered in liquid nitrogen and stored at −80 °C. These plant samples were used for gene expression analysis by qPCR.

### 4.7. RNA Extraction and RT-qPCR Analysis (Second ISR Experiment)

Prior to RNA extraction, samples were grounded to a fine powder with liquid nitrogen. Total RNA was isolated from each replicate with PureLink RNA Micro Kit (Invitrogen), DNAase treatment included. RNA purity was confirmed using NanodropTM. A retrotranscription followed by RT-qPCR was performed.

The retrotranscription was performed using iScript tm cDNA Synthesis Kit (Bio-Rad). All retrotranscriptions were carried out using a GeneAmp PCR System 2700 (Applied Biosystems): 5 min 25 °C, 30 min 42 °C, 5 min 85 °C, and hold at 4 °C. Amplification was carried out with a MiniOpticon Real Time PCR System (Bio-Rad): 3 min at 95 °C and then 39 cycles consisting of 15 s at 95 °C, 30 s at 55 °C and 30 s at 72 °C, followed by melting curve to check results. To describe the expression obtained in the analysis, cycle threshold (Ct) was used. Standard curves were calculated for each gene, and the efficiency values ranged between 90 and 110%. Results for gene expression were expressed as differential expression by the 2^−ΔΔCt^ method. *Sand* gene (AT2G28390) was used as reference gen [79]. Gene primers used are shown in Table 5.

### 4.8. Metabolic Elicitors’ Extraction and Its Capacity to Induce Systemic Resistance. Third ISR Experiment

Based on data from qPCRs (second ISR experiment), and protection from the first ISR experiment, two strains were chosen to isolate their metabolic elicitors and check their capacity to mimic bacterial protection: *S. rubidaea* N 12.34 because it was the one with best differential expression results (Figure 4) and *B. cereus* N 4.1 because it was the Gram-positive one with best protection against disease results (Table 3).

Metabolic elicitors were extracted according to Sumayo et al. [28] protocol until obtaining n-hexane, ethyl acetate and n-butanol fractions. Briefly, strains were grown in nutrient broth (CONDA) on a rotary shaker (180 rpm) at 28 °C for 24 h. Cells were eliminated by centrifugation at 8000× *g* for 15 min. Five hundred mL of the obtained supernatant was filtrated by a 0.2 µm nitrocellulose filter. This filtrate was used to extract metabolic elicitors. First, a double extraction 1:1 (*v/v*) with n-hexane was made. The remaining aqueous phase was extracted twice with ethyl acetate (1:1 *v/v*), and finally, the aqueous phase was extracted twice with n-butanol (1:1 *v/v*). The organic phases (n-hexane, ethyl acetate and n-butanol) were pooled and evaporated to dryness in a rotary evaporator at 50 °C. The dry residues obtained were dissolved in 25 mL of 10% Dimethyl sulfoxide (DMSO).

A third ISR assay on *A. thaliana* plants to evaluate the ability of the three metabolic elicitor fractions extracted from N 12.34 and N 4.1 strains was carried out. Four treatments per strain were defined: (a) Metabolic elicitors in the n-hexane fraction, (b) metabolic elicitors in the ethyl acetate fraction, (c) metabolic elicitors in the n-butanol fraction, and (e) positive control (BTH [28]). Additional controls (negative control) with the fractions extracted with n-hexane, ethyl acetate and n-butanol from nutrient broth (without bacteria) and dissolved in 10% DMSO were also included to ensure that elicitor effects were due to bacterial components and not to the nutrient broth or the DMSO. All were pathogen challenged.

*A. thaliana* was handled as described in the first ISR assay (Figure 8). Treatments were delivered to seedlings by soil drench (50 µL of the three metabolic elicitor fractions, 10 µL of BTH (positive control), and 50 µL of each negative control fraction). The pathogen was also inoculated as described in the first ISR assay. Seventy-two hours after pathogen inoculation, disease severity was recorded and relativized as in the first ISR experiment (Figure 8).

### 4.9. RT-qPCR Analysis of the Genes Triggered by Metabolic Elicitor Fractions (Fourth ISR Experiment)

Based on data from the third ISR experiment, another ISR assay was carried out using the protocol explained above. The two most effective metabolic elicitor fractions against pathogen attack from each bacteria (n-hexane and ethyl acetate) were used. Differential gene expression of *NPR1*, *PR2* and *PDF1* for strain N 12.34 and *NPR1*, *PR3,* and *PDF1* for strain N 4.1 were analyzed. In the case of strain N 12.34, analysis was performed at 6 and 12 hapc, and in that of strain N 4.1, at 12 and 24 hapc. Genes and sampling moments were selected according to previous results of the first qPCR experiment.

*A. thaliana* was handled as described in the first ISR assay (Figure 8). Treatments were n-hexane metabolic elicitor fraction from N 12.34, ethyl acetate metabolic elicitor fraction from N 12.34, n-hexane metabolic elicitor fraction from N 4.1, ethyl acetate metabolic elicitor fraction from N 4.1 and controls with n-hexane and ethyl acetate (sterile nutrient broth was used to obtain control n-hexane and control ethyl-acetate fractions). Plants were inoculated by soil drench (50 µL), and challenge inoculation with *P. syringae* DC3000 was performed as explained above.

### 4.10. Statistical Analysis

One-way ANOVA with replicates was used to check the statistical differences in all data obtained. Prior to ANOVA analysis, homoscedasticity and normality of the variance was checked with Statgraphics plus 5.1 for Windows, meeting requirements for analysis. When significant differences appeared (*p* < 0.05) a Fisher test was used [80].

## 5. Conclusions

The enormous biotechnological potential of the rhizosphere as a source of bacterial strains capable of establishing a beneficial relationship with plants and of modifying their defensive metabolism, improving their ability to defend themselves from pathogen attacks, has been evidenced.

In addition, triggering SA and/or JA/ET defensive pathways by bacteria seem to be more complex than current description in the literature and the concept of simultaneous elicitation of different pathways of plant immune system has been reinforced.

Each bacterium had a different effect in the genes studied, even within the same bacterial genus. In addition, the metabolic elicitors of the two studied strains had different effects to that produced by the bacteria, confirming the presence of many different bacterial molecules able to trigger plant metabolism. This is very interesting since it opens a huge amount of biotechnological possibilities to develop biological products for agriculture in different situations and plant species

## Figures and Tables

**Figure 1 plants-09-01731-f001:**
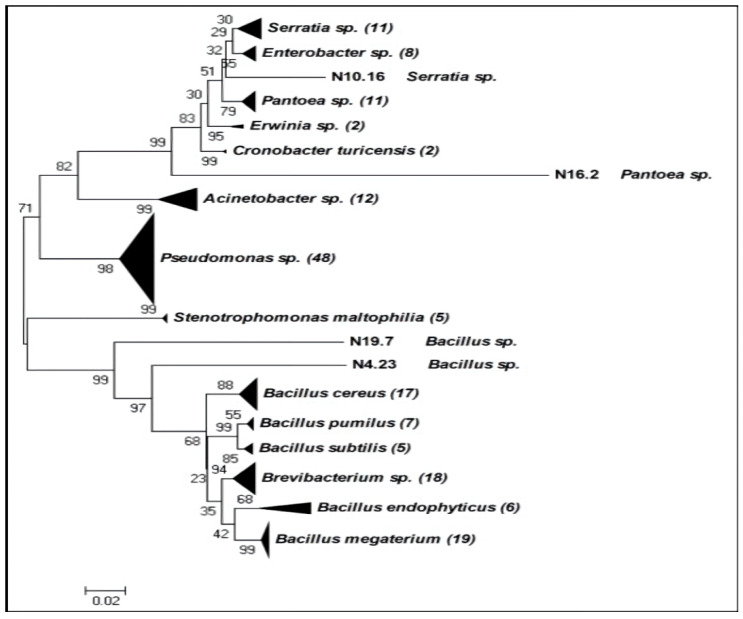
Phylogenetic tree performed with the 16S rRNA sequences. The evolutionary distances were inferred using the Neighbor-Joining method. The bootstrap consensus tree inferred from 1000 replicates was taken to represent the evolutionary history of the taxa analyzed. Annotation of bacteria included in the tree, were obtained from the NCBI (http://www.ncbi.nlm.nih.gov/). The number in brackets indicates number of species within each phylogenetic group.

**Figure 2 plants-09-01731-f002:**
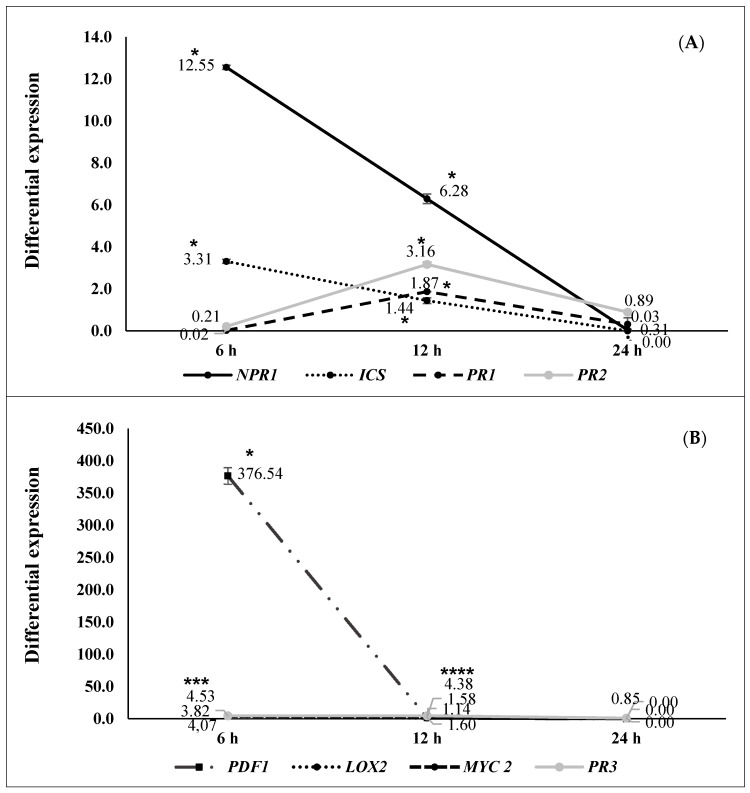
Differential gene expression (seedlings inoculated with N 5.12 (*Pseudomonas putida*) vs negative control) at 6 (*n* = 16), 12 (*n* = 16) and 24 (*n* = 16) hapc; (**A**) *NPR1*, *ICS*, *PR1,* and *PR2* genes (as SA signaling pathway markers) and (**B**) *PDF1*, *LOX2*, *MYC2,* and *PR3* (as JA/ET signaling pathway markers). Asterisks represent statistically significant differences (*p* < 0.05) with respect to negative control (differential expression of 1).

**Figure 3 plants-09-01731-f003:**
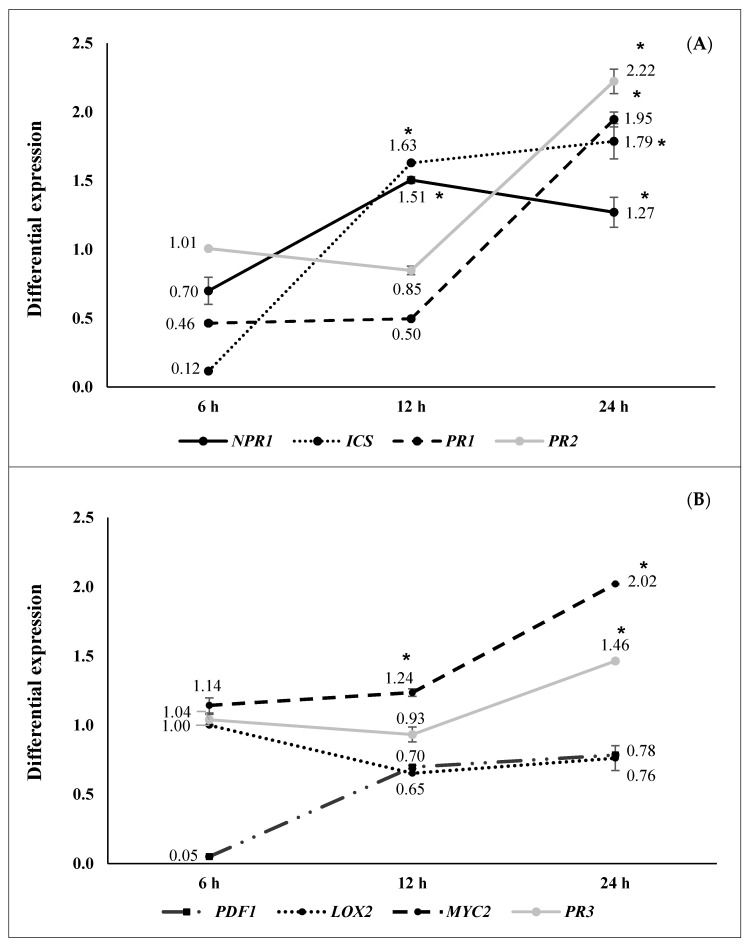
Differential gene expression (seedlings inoculated with N 8.17 (*Stenotrophomonas maltophilia*) vs negative control) at 6 (*n* = 16), 12 (*n* = 16), and 24 (*n* = 16) hapc; (**A**) *NPR1*, *ICS*, *PR1,* and *PR2* genes (as SA signaling pathway markers) and (**B**) *PDF1*, *LOX2*, *MYC2,* and *PR3* (as JA/ET signaling pathway markers). Asterisks represent statistically significant differences (*p* < 0.05) with respect to negative control (differential expression of 1).

**Figure 4 plants-09-01731-f004:**
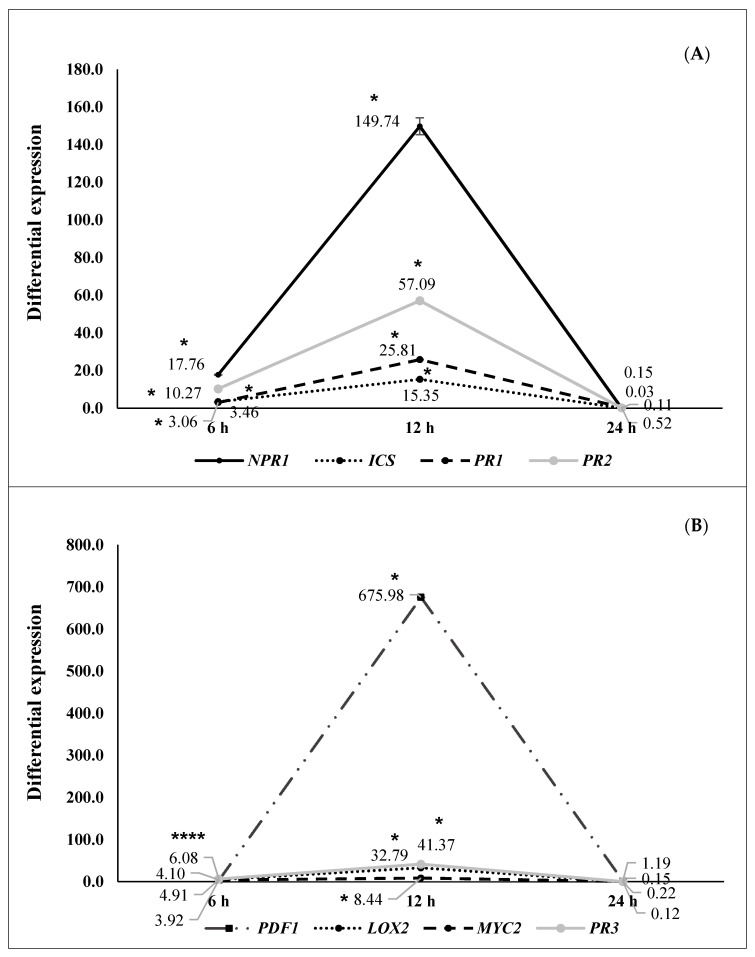
Differential gene expression (seedlings inoculated with N 12.34 (*Serratia rubidaea*) vs negative control) at 6 (*n* = 16), 12 (*n* = 16), and 24 (*n* = 16) hapc; (**A**) *NPR1*, *ICS*, *PR1,* and *PR2* genes (as SA signaling pathway markers) and (**B**) *PDF1*, *LOX2*, *MYC2,* and *PR3* (as JA/ET signaling pathway markers). Asterisks represent statistically significant differences (*p* < 0.05) with respect to negative control (differential expression of 1).

**Figure 5 plants-09-01731-f005:**
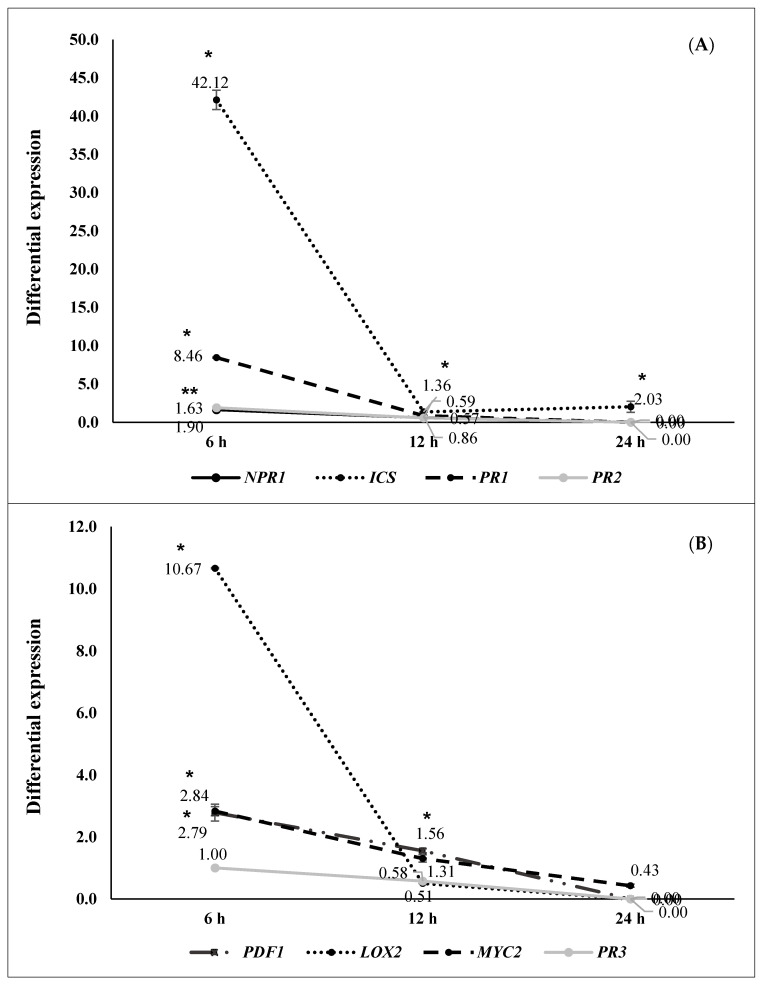
Differential gene expression (seedlings inoculated with N 21.24 (*Pseudomonas fluorescens*) vs negative control) at 6 (*n* = 16), 12 (*n* = 16) and 24 (*n* = 16) hapc; (**A**) *NPR1*, *ICS*, *PR1,* and *PR2* genes (as SA signaling pathway markers) and (**B**) *PDF1*, *LOX2*, *MYC2,* and PR3 (as JA/ET signaling pathway markers). Asterisks represent statistically significant differences (*p* < 0.05) with respect to negative control (differential expression of 1).

**Figure 6 plants-09-01731-f006:**
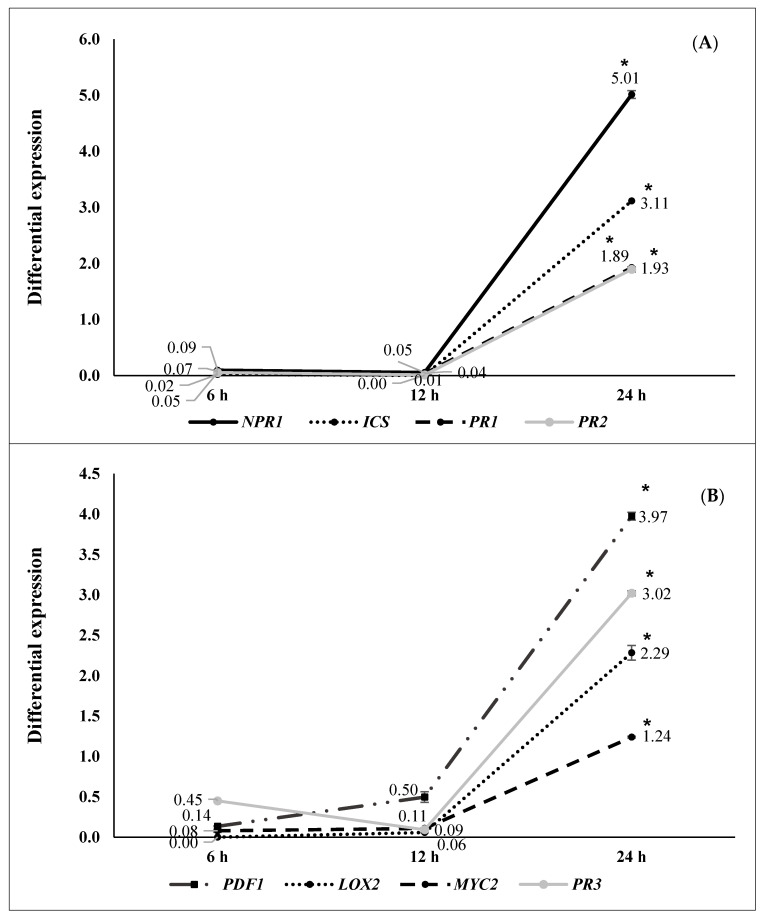
Differential gene expression (seedlings inoculated with N 4.1 (*Bacillus cereus*) vs negative control) at 6 (*n* = 16), 12 (*n* = 16) and 24 (*n* = 16) hapc; (**A**) *NPR1*, *ICS*, *PR1,* and *PR2* genes (as SA signaling pathway markers) and (**B**) *PDF1*, *LOX2*, *MYC2,* and *PR3* (as JA/ET signaling pathway markers). Asterisks represent statistically significant differences (*p* < 0.05) with respect to negative control (differential expression of 1).

**Figure 7 plants-09-01731-f007:**
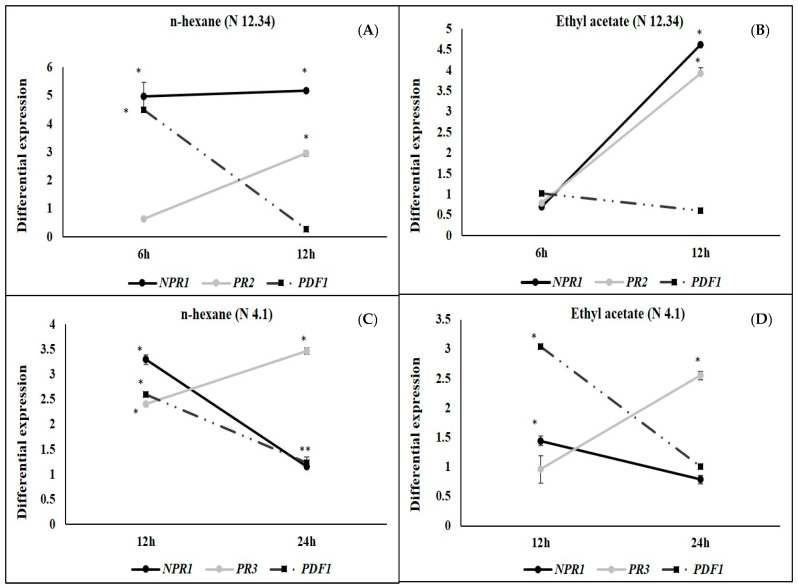
Differential gene expression in plants under the following treatments: (**A**) N 12.34 elicitors from the n-hexane fraction; (**B**) N 12.34 elicitors from the ethyl-acetate fraction; (**C**) N 4.1 elicitors from the n-hexane fraction; and (**D**) N 4.1 elicitors from the ethyl-acetate fraction vs negative control, at 6 (*n* = 16) and 12 (*n* = 16) hapc in N 12.34 (**A**,**B**), and at 12 (*n* = 16) and 24 (*n* = 16) hapc in N 4.1 (**C**,**D**). *NPR1* and *PR2* genes as markers of the SA signaling pathway and *PDF1* as marker of the JA/ET signaling pathway in N 12.34; *NPR1* as marker of the SA signaling pathway, and *PDF1* and *PR3* as markers of the JA/ET signaling pathway in N 4.1. Asterisks represent statistically significant differences (*p* < 0.05) within each sampling time. Genes and sampling times were chosen based on results obtained by bacterial strains (Figure 4 and Figure 6).

**Figure 8 plants-09-01731-f008:**
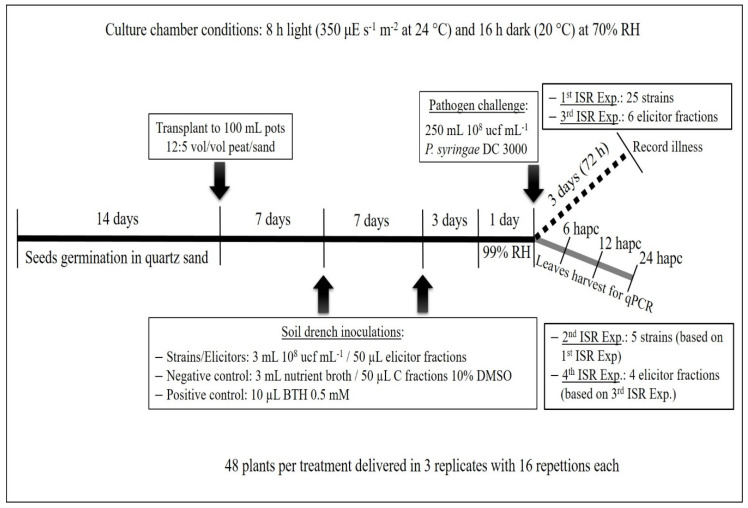
Induced systemic resistance (ISR) experiments represented as a timeline. The black continuous line represents the part of the experimental design common to all ISR experiments; the black dashed line represents the last part of ISR experiments to assess protection against the pathogen (first and third experiments) and the grey continuous line represents the last part of ISR experiments to carry out differential gene expression analyses by qPCR (second and fourth experiments).

**Table 1 plants-09-01731-t001:** Percentage of bacteria within each genera or species (in Gram-positive group), positive for biochemical traits.

	**Gram-Negative Group**							
**Biochemical Traits**	*Serratia*	*Enterobacter*	*Pantoea*	*Erwinia*	*Cronobacter*	*Acinetobacter*	*Pseudomonas*	*Stenotrophomonas*
Indole acetic acid (IAA) production	0.00	37.5	0.00	0.00	0.00	0.00	0.00	0.00
Siderophores production	100.00	87.5	100.00	100.00	100.00	83.33	100.00	100.00
Phosphate solubilization	0.00	75.00	45.45	50.00	0.00	100.00	91.67	0.00
Chitinases production	0.00	0.00	0.00	0.00	0,00	0.00	2.08	100.00
Siderophores production and phosphate solubilization	0.00	62.50	45.45	50.00	0.00	83.33	85.42	0.00
Siderophores and chitinases production	0.00	0.00	0.00	0.00	0.00	0.00	2.08	100.00
Siderophores and IAA production and phosphate solubilization	0.00	25.00	0.00	0.00	0.00	0.00	0.00	0.00
	**Gram-Positive Group**							
**Biochemical Traits**	*Bacillus cereus*	*Bacillus pumillus*	*Bacillus subtilis*	*Brevibacterium* sp.	*Bacillus endophyticus*	*Bacillus megaterium*		
IAA production	0.00	0.00	0.00	0.00	0.00	0.00		
Siderophores production	100.00	100.00	100.00	100.00	100.00	100.00		
Phosphate solubilization	23.08	0.00	0.00	11.76	0.00	36.84		
Chitinases production	46.15	0.00	20.00	0.00	0.00	0.00		
Siderophores production and phosphate solubilization	23.08	0.00	0.00	11.76	0.00	36.84		
Siderophores and chitinases production	15.38	0.00	20.00	0.00	0.00	0.00		
Siderophores and chitinases production and phosphate solubilization	7.69	0.00	0.00	0.00	0.00	0.00		

Biochemical traits are indole acetic acid (IAA) production, siderophores production, phosphate solubilization, chitinases production and the combination of these traits.

**Table 2 plants-09-01731-t002:** Twenty-five selected strains and its biochemical traits.

			Biochemical Traits			
	Bacterial Strain		IAA Production	Siderophores Production	Chitinases Production	Phosphate Solubilization
**GRAM−**	N 5.12	*Pseudomonas putida*		+		+
	N 8.17	*Stenotrophomonas maltophilia*		+	+	
	N 8.22	*Stenotrophomonas* sp.		+	+	
	N 9.11	*Pseudomonas reinekei*		+		+
	N 10.6	*Pseudomonas putida*		+		+
	N 10.7	*Serratia odorifera*		+		
	N 10.21	*Pseudomonas putida*		+		+
	N 12.34	*Serratia rubidaea*		+		
	N 15.23	*Pseudomonas brassicacearum*		+		+
	N 16.3	*Pantoea* sp.		+		+
	N 16.15	*Enterobacter* sp.	+			+
	N 16.23	*Pantoea agglomerans*		+		+
	N 16.24	*Enterobacter sp.*	+	+		+
	N 18.10	*Pseudomonas fragi*		+		+
	N 21.24	*Pseudomonas fluorescens*		+		+
**GRAM+**	N 4.1	*Bacillus cereus*		+	+	
	N 5.20	*Bacillus cereus*		+	+	+
	N 8.10	*Bacillus* sp.		+		+
	N 11.5	*Brevibacterium* sp.		+		+
	N 11.14	*Bacillus endophyticus*		+		
	N 11.20	*Bacillus atrophaeus*		+	+	
	N 11.22	*Bacillus megaterium*		+		+
	N 11.36	*Bacillus megaterium*		+		+
	N 11.40	*Bacillus megaterium*		+		+
	N 20.15	*Bacillus simplex*		+		+

Biochemical traits are indole acetic acid (IAA) production, siderophores production, phosphate solubilization and chitinases production. A positive biochemical trait of each bacteria is indicated by a + symbol.

**Table 3 plants-09-01731-t003:** Percentage of protection (%) induced in *A. thaliana* seedlings inoculated with chosen strains against DC3000.

Treatment			% of Protection
**Controls**	Negative Control	Nutrient Broth	0
	Positive Control	Benzothiadiazole (BTH)	54.21 ± 4.03 *
**GRAM− Strains**	**N 5.12**	***Pseudomonas putida***	**57.69 ± 1.76 ***
	**N 8.17**	***Stenotrophomonas maltophilia***	**64.87 ± 1.79 ***
	N 8.22	*Stenotrophomonas* sp.	0
	N 9.11 N 10.6	*Pseudomonas reinekei* *Pseudomonas putida*	51.44 ± 6.88 *
			0
	N 10.7 N 10.21	*Serratia odorífera* *Pseudomonas putida*	33.76 ± 3.22 * 0
	**N 12.34**	***Serratia rubidaea***	**56.64 ± 2.15 ***
	N 15.23	*Pseudomonas brassicacearum*	0
	N 16.3	*Pantoea* sp.	14.91 ± 2.45 *
	N 16.15	*Enterobacter* sp.	24.18 ± 1.96 *
	N 16.23	*Pantoea agglomerans*	21.21 ± 7.32 *
	N 16.24	*Enterobacter sp.*	6.93 ± 2.31
	N 18.10	*Pseudomonas fragi*	0
	**N 21.24**	***Pseudomonas fluorescens***	**82.08 ± 2.46 ***
**GRAM+ Strains**	**N 4.1**	***Bacillus cereus***	**69.45 ± 0.38 ***
	N 5.20	*Bacillus cereus*	49.75 ± 0.82 *
	N 8.10	*Bacillus* sp.	22.93 ± 2.93 *
	N 11.5	*Brevibacterium* sp.	29.82 ± 1.82 *
	N 11.14	*Bacillus endophyticus*	0
	N 11.20	*Bacillus atrophaeus*	42.72 ± 3.51 *
	N 11.22	*Bacillus megaterium*	0
	N 11.36	*Bacillus megaterium*	0
	N 11.40	*Bacillus megaterium*	23.98 ± 0.18 *
	N 20.15	*Bacillus simplex*	30.83 ± 4.92 *

The percentage of protection was calculated based on the number of leaves with disease symptoms to the total of leaves (*n* = 12 seedlings per replicate). Data were relativized to negative control (seedlings inoculated only with nutrient broth and pathogen challenged), which was considered as 0% protection. A positive control (BTH) was also used [28]. Strains in bold are those whose percentage of protection against the pathogen *P. syringae* DC3000 exceeded that of the positive control and therefore, those that were selected for further analyses. Asterisks represent statistically significant differences (*p* < 0.05) with regard to negative control.

**Table 4 plants-09-01731-t004:** Percentage of protection (%) induced in *A. thaliana* seedlings inoculated with the elicitor fractions against the pathogen *P. syringae* DC3000.

Treatment		% of Protection
**Controls**	Negative control	0
	Positive control (BTH)	52.09 ± 1.75 *
**N 12.34**	**n-Hexane**	**61.26 ± 2.23 ***
	**Ethyl acetate**	**54.64 ± 1.48 ***
	n-Butanol	35.42 ± 2.77 *
**N 4.1**	**n-Hexane**	**68.11 ± 0.76 ***
	**Ethyl acetate**	**67.30 ± 3.76 ***
	n-Butanol	52.31 ± 1.91 *

*A. thaliana* seedlings were elicited with the n-hexane, ethyl acetate and n-butanol fractions extracted from strains N 12.34 and N 4.1. Percentages were estimated according to the number of leaves with pathogen infection symptoms with respect to the total of leaves (*n* = 16 seedlings per replicate). Negative control was considered as 0% of protection and then data were relativized with respect to it. A positive control (BTH) was also used [28]. Fractions in bold are those whose percentage of protection against the pathogen exceeded that of the positive control and therefore, those that were selected for further analyses. Asterisks indicate statistically significant differences (*p* < 0.05) with respect to negative control.

**Table 5 plants-09-01731-t005:** Forward and reverse primers used in qPCR analysis.

	Forward Primer	Reverse Primer
***AtNPR1***	5′-TATTGTCAARTCTRATGTAGAT	5′-TATTGTCAARTCTRATGTAGAT
***AtPR1***	5′-AGTTGTTTGGAGAAAGTCAG	5′-GTTCACATAATTCCCACGA
***AtICS***	5′-GCAAGAATCATGTTCCTACC	5′AATTATCCTGCTGTTACGAG
***AtPdf1***	5′-TTGTTCTCTTTGCTGCTTTCGA	5′-TTGGCTTCTCGCACAACTTCT
***AtLOX2***	5′-ACTTGCTCGTCCGGTAATTGG	5′-GTACGGCCTTGCCTGTGAATG
***AtMYC2***	5′GATGAGGAGGTGACGGATACGGAA	5′-CGCTTTACCAGCTAATCCCGCA
***AtPR2***	5′-TCGTCTCGATTATGCTCTCTTC	5′-GCAGAATACACAGCATCCAAAA
***AtPR3***	5′-AAATCAACCTAGCAGGCCACT	5′-GAGGGAGAGGAACACCTTGACT
***Sand***	5′ -CTGTCTTCTCATCTCTTGTC	5′-TCTTGCAATATGGTTCCTG

*At = A. thaliana*.

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
