# Peer review of "Bioeffectors as Biotechnological Tools to Boost Plant Innate Immunity: Signal Transduction Pathways Involved"

_plants, 2020, doi:10.3390/plants9121731_

Round 1
Reviewer 1 Report
-The paper is interesting and well performed and only minor point should be addressed.
-In the introduction please specify better why N. glauca was chosen for this study.
-Give also information on why analyses were performed in A. thaliana.
-Additional information on the biochemical traits selected for the screening should be given
-Why the section materials and method was between results and discussion?
Author Response
First of all, we would like to thank you for your constructive criticism and the effort made in the thorough correction that has improved the manuscript. We really appreciate the deep revision and correction.
We have accepted all the changes and suggestions proposed and the manuscript has improved the quality.
The changes we have made appear in italics after the reviewer’s suggestions:
- In the introduction, please specify better why glauca was chosen for this study.
As the reviewer appropriately suggests, in the introduction we have expanded the information requested on why we have chosen the N. glauca plant for our study. This information is expanded in lines 65-78.
“It is known that the rhizosphere of wild plant species is a worthy source for finding putative effective beneficial bacteria (bioeffectors) because plants are able to select those bacteria that boost their fitness by releasing nutrients into the surrounding soil through root exudates [18, 19]. Thanks to this well-known ability of the plants to strongly select beneficial bacterial strains in the rhizosphere to improve their health and to survive to adverse conditions [18-20], bacteria isolated from the rhizosphere of Nicotiana glauca Graham, a Solanaceae native of Southern Spain with a strong secondary metabolism [21], were studied. This plant was chosen for our work because it is a wild species capable of thriving in very poor soils and in extreme drought and temperature conditions, therefore it was deduced that it should have a powerful associated rhizospheric microbiome that would allow it to survive to these harsh conditions. Moreover, N. glauca synthesizes anabasin, a toxic alkaloid, which suggests the existence of an inducible secondary metabolism associated to defence mechanisms [21]. All these characteristics of this plant species led us to presumed that the rhizosphere of wild populations of N. glauca would be a rich source for finding effective bioeffectors with very interesting metabolic capacities.”
From the sampling carried out with the rhizosphere of N. glauca, the use (as plant inoculant) of one of the isolated rhizobacterium were patented due to its effectiveness in boosting plant tolerance to biotic and abiotic stress. Another patent with another rhizobacterium is also being performed for its ability to enhance flavonoid metabolism in blackberry cultivars. This demonstrates the hypothesis that the rhizosphere of N. glauca is an extraordinary good source for the extraction of beneficial rhizobacteria (bioeffectors).
- Give also information on why analyses were performed in thaliana.
As the reviewer appropriately suggests, in the introduction we have expanded the information requested on why we have chosen the A. thaliana plant for our experiments. This information is expanded in lines 96-102.
“The ISR experiments were carried out in A. thaliana plants because its short life cycle facilitates the performance of experiments at laboratory level. Furthermore, it is easy to infect this plant with specific pathogens (e.g. P. syringae DC 3000) and to study the consequent plant immune responses. This plant is able to trigger a wide sort of general PTI and ETI responses against pathogens (Cui et al. 2014). In addition, it exists extensive literature on the defensive responses of this plant to compare with our results. As it is a model plant, the data obtained would be very relevant for the scientific community and could be then extrapolated to crops of commercial or economic interest.”
The elucidation of these immune response pathways and of the regulatory networks of A. thaliana plants has extensively contributed to the general understanding of plant defences against pathogens and to comprehend the establishment of beneficial interactions with microorganisms. This has also led to the discovery of the genetic basis of agronomically essential traits in crops.
- Additional information on the biochemical traits selected for the screening should be given.
As suggested by the reviewer, the information regarding to the biochemical traits selected for the screening has been expanded in the material and methods section (lines 446-467).
“For the detection of auxin-like substances, a colorimetric technique was used. Bacterial isolates were inoculated onto half-strength Tryptic Soy Agar (TSA; Difco Laboratories, Sparks,MD) in a grid pattern. Each inoculated plate was overlaid with an 82-mm-diameter nitrocellulose membrane (Amersham Biosciences) and subsequently, Salkowski’s reagent (2% (v/v) 0.5 M FeCl3 in 35% (v/v) perchloric acid) was applied to the membrane and incubated for 2 h to allow for colour reaction development. Sensitivity of the assay was determined using known concentrations of pure IAA (Sigma Chemical Co.).
To detect the production of siderophores, the culture medium described by Alexander and Zuberer [24] was used. This medium contains CAS (Chrome azurol S), to which iron is bound as part of the Fe-CAS complex, blue in color; those bacteria capable of producing siderophores separate the iron from the complex producing a yellow halo around the bacterial growth zone, a halo that is measured in mm after 24 hours of incubation at 28 ºC.
To demonstrate the ability of the strains to solubilize phosphate, the bacteria were cultured in the medium described by De Freitas et al. [25], which contains potato dextrose agar and yeast extract (PDYA) with freshly precipitated calcium phosphate. The hydrolysis halo made in this culture medium around the colonies was measured in mm after 24 h of incubation at 28 ºC. The presence of a hydrolysis halo confirms the ability of the strain to solubilize phosphate.
To assess the ability of the strains to produce chitinases, the culture medium described by Frändberg and Shunürer et al. [27] was used. This medium contains colloidal chitin, K2HPO4, KH2PO4, MgSO4, NaCl, KCl, yeast extract and agar. Those bacteria able to produce chitinases produce a transparent halo (the medium is opaque) around the bacterial growth zone, a halo that is measured in mm after 5 days of incubation at 30 ºC.”
- Why the section materials and method was between results and discussion?
Putting the materials and methods section between the results and the discussion has been a confusion and has been corrected, placing the materials and methods section after the discussion, as suggested by the “instructions for authors” section of the journal.

Reviewer 2 Report
The authors reported article: Bioeffectors as biotechnological tools to boost plant innate immunity: signal transduction pathways involved.
The chosen topic is very relevant all over the world for a comprehensive study of widely used plants as raw materials and its healthy. The authors of the abstract have very thoroughly described the topicality of the topic from all aspects, problems related to this topic. about plant immune system. The aims of the work are very concise.
The experimental section describes the course of the experiments in great detail, allowing you to track the results up to. Statistical processing of data has been performed.
The results are widely described and comprehensively evaluated and the main conclusions are given for those who like to understand the truthfulness of the obtained results.
Author Response
Reviewer 2 does not suggest any changes or modifications to the manuscript.
We would like to thank you for the effort made in the thorough correction of the manuscript. We really appreciate the deep revision.